

# Soil organic matter interactions along the elevation gradient of the James Ross Island (Antarctica)

Vítězslav Vlček[1], David Juřička[2], Martin Valtera[3], Helena Dvořáčková[1], Vojtěch Štulc[1], Michaela Bednaříková[4], Jana Šimečková[1], Peter Váczi[4], Miroslav Pohanka[6], Pavel Kapler[5], Miloš Barták[4], Vojtěch Enev[7].

[1] Mendel University in Brno, Faculty of AgriSciences, Department of Agrochemistry, Soil Science, Microbiology and Plant Nutrition, Zemědělská 1, 613 00 Brno, Czech Republic.
[2] Mendel University in Brno, Faculty of Forestry and wood technology, Department of Geology and Pedology, Zemědělská 3, 613 00 Brno, Czech Republic.
[3] Mendel University in Brno, Faculty of Forestry and wood technology, Department of Forest Botany, Dendrology and Geobiocoenology, Zemědělská 3, 613 00 Brno, Czech Republic.
[4] Masaryk University, Faculty of Science, Department of Experimental Biology, Kamenice 5, 62500 Brno, Czech Republic.
[5] Masaryk University, Faculty of Science, Department of Geography, Kotlářská 2, 611 37 Brno, Czech Republic.
[6] University of Defence, Faculty of Military Health Sciences, Třebešská 1575, CZ-500 01 Hradec Králové, Czech Republic.
[7] Brno University of Technology, Faculty of Chemistry, Institute of Physical and Applied Chemistry, Purkynova 118, 612 00 Brno, Czech Republic.

*Correspondence to*: Vítězslav Vlček (vitezslav.vlcek@mendelu.cz)

**Abstract.** Around half of the Earth's soil organic carbon (SOC) is presently stored in the Northern permafrost region. In polar permafrost regions, low temperatures particularly inhibit both the production and biodegradation of organic matter. In such conditions, abiotic factors such as mesoclimate, pedogenic substrate or altitude are thought to be more important for soil development than biological factors. In Antarctica, biological factors are generally underestimated in soil development due to the rare occurrence of higher plants and the short time since deglaciation. In this study, we aim to assess the relationship between SOC and other soil properties related to the pedogenic factors or properties. Nine plots were investigated along the altitudinal gradient from 10 to 320m at the deglaciated area of James Ross Island (Ulu Peninsula) with a parallel tea-bag decomposition experiment. SOC contents showed a positive correlation with the contents of easily extractable glomalin-related soil proteins (EE-GRSP; Spearman r = 0.733, $P$=0.031) and the soil buffer capacity (expressed as a $\Delta$pH; Spearman r = 0.817, $P$=0.011). The soil available P was negatively correlated with altitude (Spearman r = -0.711, $P$=0.032) and the exchangeable Mg was negatively correlated to the content of rock fragments (Spearman r = -0.683, $P$=0.050)No correlation was found between the available mineral nutrients (P, K, Ca, Mg) and SOC nor GRSP. This may be a consequence of the inhibition of biologically mediated nutrient cycling in the soil. Therefore, the main factor influencing nutrient availability in these soils seems to be not the biotic, but rather the abiotic environment influencing the mesoclimate (altitude) or the level of weathering (rock content). Incubation in tea bags for 45 days resulted in the consumption and/or translocation of more labile polyphenolic and water-extractable organic matter, along with changes in C content (increase of up to +0.53% or decrease of up to -1.31% C) and a decrease in the C:N ratio (from 12.5 to 7.1-10.2), probably due to microbial respiration and an increase in the abundance of nitrogen binding microorganisms. Our findings suggest that one of the main variables influencing SOC/GRSP





content is not altitude or coarse fraction content (whose correlation with SOC/GRSP were not found) but probably other factors that are difficult to quantify, such as the availability of liquid water.

## 1 Introduction

The oceans and underground fossil resources are the two largest carbon 'pools' within the global carbon cycle, while topsoil,
with an estimated 2,000 + petagrams (Pg-C) of sequestered carbon, is considered the third greatest carbon sink. Most of this carbon (2200Pg-C) is in the form of soil organic matter (SOM), with a limited amount in the form of carbonates (Batjes, 1996). In addition, a further 600 Pg-C is stored in the form of vegetation, while ca. 780 Pg-C is stored in the atmosphere (Rackley, 2017).

Soil carbon dynamics are mainly influenced by climatic factors and soil properties, which determine both the stability and
decomposition rate of SOM (Aerts, 1997). Overall, the annual level of soil carbon flux, i.e. the amount of carbon emitted from the soil to the atmosphere due to microorganisms and root respiration, is estimated to be between 70–100 Pg-C (Bond-Lamberty & Thomson, 2010). An important factor in this global carbon cycle flux is SOM mineralisation, which contributes ca. 58 Pg-C annually to the total (Zech et al. 1997; Houghton, 2007). On a local scale, other factors will also play a role in the cycling of carbon, such as the nature of the organic matter, local topography and composition of local vegetation and the
microbial community (Makkonen et al., 2012, Bonanomi et al., 2013). Most of these factors, however, are directly or indirectly related to (micro)climatic conditions (Röder et al., 2016; Wilcke et al., 2008).

Currently, extensive research is being undertaken on the impact of climate change on terrestrial ecosystems, with permafrost areas seen as amongst the most vulnerable (Masson-Delmotte et al. 2006; Durán et al. 2021; Swati et al. 2021). The cold conditions prevalent at high latitudes are assumed to stunt the biological decomposition of organic matter, leading to the
formation of soils rich in organic matter. This is especially true in northern hemisphere permafrost regions, where it is estimated that around half of the Earth's soil organic carbon (SOC) is presently stored (Tarnocai et al., 2009). Under Antarctic conditions, however, soil formation processes are thought to be affected more by climate, soil substrate and topography (e.g. altitude) than biological factors, due both to a general absence of higher vegetation and the relatively short period for pedogenesis (soil formation) since deglaciation. Consequently, soils in the Antarctic region tend to have a very low SOM content (Vlček et al.
2018; Pospíšilová et al. 2017; Zvěřina et al. 2012) with rare occurrence of organic horizons (Campbell and Claridge, 1987).

Glomalin, or more commonly 'glomalin-related soil protein' (GRSP; a crude extract containing many substances, including humic acids), is a hypothetical glycoprotein produced abundantly on the hyphae and spores of arbuscular mycorrhizal fungi (AMF; Glomeromycota) in the soil and roots and, as such, is an important component in SOM (Schüßler et al., 2001). Though there is some debate over its origin (e.g. see Gillespie et al. 2011; Holátko et al. 2021), GRSP is thought to be a sequential
homologue of the chaperone protein group, involved in the folding of primary proteins, widespread across all domains of life, after their translation on ribosomes (Alaei et al., 2021). Arbuscular mycorrhiza is the most widespread form of mutualism between photobionts and fungi, represented in 70–90 % of vascular plant species (Basu et al., 2018; Błaszkowski, 2012; Fitter



et al., 2008; Smith & Read, 2008). Fossil records indicate that arbuscular mycorrhiza has played an important role in terrestrial ecosystems for as long as 250–400 million years (Harper, et al. 2013; Redecker et al., 2000; Remy et al., 1994), with examples

present in the earliest stages of the plant world's colonisation of terrestrial ecosystems (Blackwell, 2000; Pirozynski & Malloch, 1975; Simon et al., 1993).

The main factors determining the presence and growth of AMF appear to be soil properties and vegetation composition (Melo et al 2019). Despite the extreme conditions found in Antarctica, non-lichenised fungi are found in the soil (Bridge & Newsham, 2009; Arenz et al., 2014) alongside lichen, which can produce allelochemicals that affect AMF (Martins et al., 2010; Tigre et

al., 2012) by influencing mycorrhizal associations and acting as mutualism stimulators (Santiágo et al., 2018). Furthermore, some organic compounds produced by lichens, such as usnic and perlatolic acid, are also important carbon sources for the soil microbial community (Stark a Hyvärinen 2003). Thus, it can be assumed that lichens and fungi are ecosystem drivers, especially in polar regions eith few other organic compound producers. Alongside the fungi and lichens, mosses, the simplest terrestrial plants, also act as photobionts, thereby playing an important role in SOC dynamics and nutrient cycling, as well as

maintaining soil structure, stability and water retention (Lovato et al. 1995; Smith & Read 1997). Numerous previous studies have confirmed symbiotic relationships between AMF and mosses (e.g. Rayner 1927; Kelley 1950; Russell & Bulman 2005). In recent years, the Antarctic (Trinity) Peninsula has been undergoing unprecedented deglaciation due to significant local warming, which has been reported as to be the most rapid on the planet over the last 50 years (Turner et al., 2005). From this perspective, the Antarctic Peninsula region represents a unique research location for early monitoring of the future

development of soils in polar regions.

The main objectives of this study were (i) to determine the relationship between GRSP, SOC, SOM and environmental conditions (e.g. altitude, content of coarse soil fraction, availability of liquid water), and (ii) to assess the influence of environmental conditions on GRSP production and SOC content. We hypothesise that the highest GRSP and SOC content will be found in soils located at the lowest elevations, along with the lowest coarse fraction content (H1). We further hypothesise

that GRSP content will be negatively correlated with the content of exchangeable phosphorous (P) (H2) and that SOC will be positively correlated with GRSP content (H3). In addition, we will (iii) monitor the initial stages of decomposition of a 'standardised organic matter sample' (tea bag) using thermogravimetric analysis (TGA). We hypothesise a more rapid OM decomposition rate associated with increased soil SOC and GRSP content, and that this will be indicative of environments with increased microbial activity (H4). Our results will contribute to improving our understanding of the processes underlying

carbon deposition in (sub-)polar Antarctic areas under ongoing climate change.





## 2. Material and methods

### 2.1 Study area

In this study, we focus on soil formation processes near the Johann Gregor Mendel Czech Antarctic Research Station in James Ross Island, one of the largest islands off the north-eastern tip of the Antarctic Peninsula. The island has an area of approx. 2 600km², of which 80% is covered with permanent glaciers. We selected nine study sites on a 300 km² deglaciated area on the Ulu Peninsula in the northern part of the island, an area covered with valleys and dome glaciers (Fig. 1). The climate is cold, with a 10-year mean annual temperature of -7.3°C along the coastal area near the Research Station, and -8°C at Bibby Hill (375 m n. m.) further inland (Matějka et al., 2021), with temperature inversions typically occurring during winter. According to Láska & Prošek (2013), elevation has little influence on temperature amplitude. Annual snow precipitation reaches 200 mm

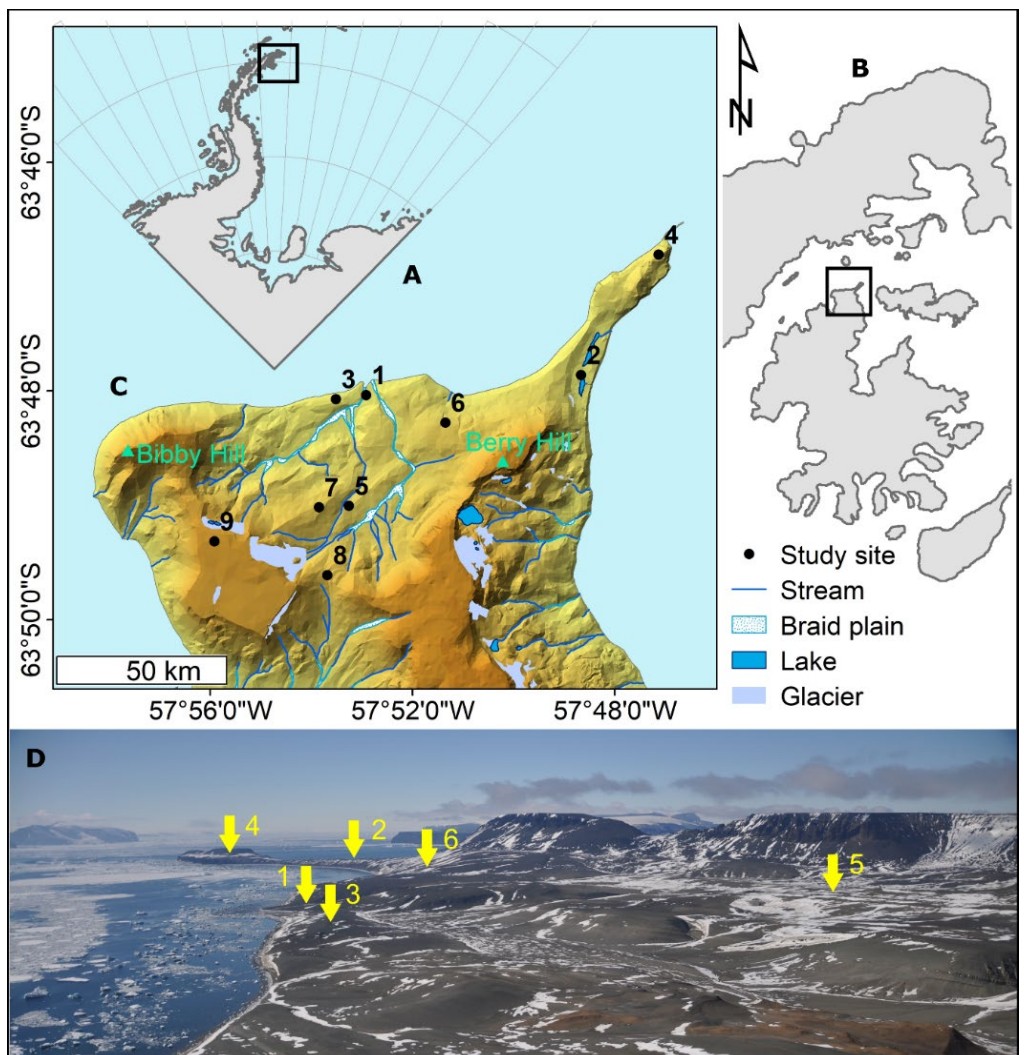



water equivalent (Strelin & Sone, 1998). Considering that the snow was subsequently redistributed by the wind, the occurrence of liquid water is more a matter of micro-relief.

**Figure 1: Situational plan of (A) the northernmost tip of the Antarctic Peninsula, with (B) James Ross Island (inset in A), (C) a digital terrain model (DTM) of the north coast of James Ross Island (Ulu Peninsula), and (D) a panoramic image of (C) with study sites 1–6 indicated (yellow arrows; view from Bibby Hill). Map created in ArcGIS Desktop**
**v.10.8.1, DTM and map layers generated using the Czech Geological Survey ČGS 2009 dataset, situational plan created in Inkscape 1.2.2. Photograph: archive of first author.**

## 2.2 Site description

The nine study sites were selected to cover a topographic gradient from lowest to highest elevation (Table 1). Site 1, known as the 'Long-Term Research Plot' (see Barták et al. 2015), was located close to the Johann Gregor Mendel Czech Antarctic
Station. The site consists of two 'plots', one open to the elements (Plot 1a) and the second in an open-topped enclosure (OTE; Plot 1b; Table 1). The land surface is covered by the marsh bryum moss *Bryum pseudotriquetrum* Hedw., which forms thick carpets interspersed with patches of lichen, such as *Rhizoplaca melanophthalma* (DC.) Leuckert & Poelt and the elegant sunburst lichen *Rusavskia elegans* (Link) S.Y. Kondr & Kärnefelt (2003). The area is rich in microbial mats formed of *Nostoc* sp. colonies and cyanobacterial species (e.g. *Microcoleus* sp.), found in seepages. Site 2 was situated between the Large and
Small Lachman Lakes and consisted of two plots divided by soil moisture conditions (dryer Plot 2a and waterlogged Plot 2b). The vegetation here is dominated by the lichens *R. elegans*, *Dermatocarpon polyphyllizum* (Korber) Hasse, *Hypnum revolutum* (Mitt.) Lindb., *Bryum argenteum* Hedw. and *Schistidium halinae* Ochyra. The site has a liquid water supply during the short summer period, though this quickly freezes as temperatures drop. Site 3, situated near the research station, was also split into two, with one part close to a seal cadaver (Plot 3a) and the second further away in an unaffected area (Plot 3b), both sites being
without vegetation cover. Site 4 was located on the side of the hill forming the Lachman Peninsula. Owing to its proximity of the shoreline, and the annual availability of water from snow melt, the site was classed as semi-moist to moist. Sites supplied with snow melt water have a layer of vegetation comprised of mosses such as *H. revolutum*, *B. argenteum* and *S. halinae*. Site 5 was located on a slope above a rivulet dubbed "the Dirty Stream", and had a surface covered with the mosses *H. revolutum* and *B. argenteum*. Owing to the annual availability of water from snow melting from the western face of the Johnson Messa
foothills, the site was classed as medium-moist. Site 6 was located at the foot of the northern slope of Berry Hill and received water from thawing snow fields on the hill. Mosses including *H. revolutum*, *B. argenteum* and *S. halinae* (Barták et al. 2015) were concentrated in the troughs of small periodic streams. Site 7 was located at the foot of the western slope of an un-named hill between "the Bohemian Stream" and "the Dirty Stream" and is classed as dry and without vegetation. Site 8 was located in the "Crame Col pass", a flat plain between the Bibby Point massive and the Lachman Crags. Again, the site is classed as
dry and without vegetation. Site 9 was located on the top of Berry Hill, with one sub-plot inside (Plot 9b) an OTE and one outside (Plot 9a). The land surface is comprised of early-developmental stage organomineral substrate and is covered with patches of lichen (*Usnea sphacelata*, *Umbilicaria decussata*).



**Table 1: Altitude, bedrock (according to Mlčoch et al., 2020), exposure and vegetation cover at each study site. M = moss, L = lichen.**

| Site | Altitude (m) | WGS84 | Bedrock | Exposure | Vegetation cover |
|------|------|-------|---------|----------|------------------|
| 1a, 1b | 10 | 63.80126S 57.88058W | Fluvial sediment | Flat | M, L |
| 2a, 2b | 15 | 63.79875S 57.80982W | Lacustrine sediment | Flat | M, L |
| 3a, 3b | 20 | 63.80178S 57.89056W | Conglomerate with mudstone and sandstone dominated intervals | Flat | None |
| 4 | 70 | 63.78135S 57.78379W | Bedded vesicular agglomerate with scoria intercalation | NW | M, L |
| 5 | 100 | 63.81735S 57.88680W | Volcanoclastic sandstone, mudstone, subordinate conglomerate | N | M |
| 6 | 102 | 63.80543S 57.85468W | Solifluction slope sediment | N | M, L |
| 7 | 130 | 63.81751S 57.89668W | Volcanoclastic sandstone, mudstone, subordinate conglomerate | W | None |
| 8 | 179 | 63.82741S 57.89416W | Sandstone, siltstone, concretionary horizons, mudstone, accretionary lapilli | NW | None |
| 9a, 9b | 320 | 63.82225S 57.93125W | Basalt, sub-aerial lava (delta caprock) | Flat | L |


## 2.3 Soil properties

Soil samples were collected from a depth of 0–5 cm at each site at the point where the decomposition of standardised organic matter experiments was to be undertaken (see Section 2.4). Soil samples were dried immediately after collection in an oven at 60 °C. The fraction smaller than 2mm and the soil skeleton were subsequently separated. Only the fine-earth fraction was used
for further analyses.

The soil reaction (pH) was determined in either distilled water or 1 M KCl at a ratio of 1:2.5 and measured on 914 Metrohm pH Meter/Conductometer. Available nutrients (P, K, Ca, Mg) were extracted using Mehlich III reagent (Mehlich A., 1984) and then assessed using an Agilant 55B AA atomic emission spectroscope (Agilent, CA, USA). For a full description of the TGA and derivative thermogravimetric (DTG) analysis of soil, see section 2.4.1.




### 2.3.1 Glomalin (GRSP) determination

Easily-extractable glomalin-related soil protein (EE-GRSP) was extracted from soil according to the procedure of Wright & Upadhyaya (1996). Briefly, 1g of soil sample was added to 8mL of 20mM trisodium citrate solution (pH 7.0). The mixture was homogenised for 30min on a GFL 3015 orbital shaker (Gesellschaft für Labortechnik mbH, Germany). The sample was

then autoclaved for 60min at 121°C and, after cooling, it was immediately centrifuged for 30min at 3 900rpm on an MPW 223e centrifuge (MPW Med. Instruments, Poland). The supernatant was then frozen at -22°C for later analysis using the method of Bradford (1976). Measurements were made in triplicate for each sample. Total glomalin (GRSP) was processed in the same manner, but instead of a 20mM trisodium citrate solution, a 50mM solution (pH 8.0) was used.

### 2.4 Standardised organic matter (tea-bag) decomposition experiment

Two four-sided tetrahedral nylon bags containing 2g (dry weight) of green tea (89% Lipton Unilever green tea (Keuskamp et al., 2013); EAN: 8714100770542) were placed into pits 5 cm below the surface at each site during the 2019 summer season and then recovered with soil. The bags had a mesh size of 0.25 mm, allowing microorganisms to enter the bags (Keuskamp et al., 2013). After 45 days of exposure to the active permafrost layer, the tea bags were removed and then dried at 60 °C, reweighed, and transported back to the Czech Republic for further analysis. Note that, though tea bags were buried at all 13

sites, only half of the samples (7 of 13) were ultimately analysed as i) some samples were destroyed by periodic freeze/thaw cycles at wetter locations, despite the test taking place in the Austral summer, and ii) some samples were taken or moved by the South Polar skua (*Stercorarius maccormicki* H. Saunders).

### 2.4.1 Thermogravimetric analysis (TGA)

Analysis of the relative element content in organic matter from the soil and tea samples was carried out through TGA analysis.

First, 0.8–1.1 mg of the tea and 8–11 mg of soil were weighed into tin capsules, which were then packaged and combusted at 980 °C using oxygen ($O_2$) injection as the combustion medium and helium as the carrier of the products formed. Relative content of nitrogen (N), carbon (C), and hydrogen (H) were evaluated based on gas chromatography determined using an EA 3000 CHNOS elemental analyser (Euro Vector, Pavia, Italy), using sulphanilamide as a standard reference for analyser calibration. The content of $O_2$ was calculated from the residual combustible mass, and the data obtained were then being

adjusted for the moisture and ash content of the samples. All analyses were performed in triplicate. Sulphur (S) content proved to be below the detection limit (0.5 wt.%) for all samples analysed.

Thermogravimetry (TG) analysis was used to assess changes in the mass of the sample over time as temperature changes. For the soil samples, 5–15 mg of the sample was weighed onto a platinum sample pan and placed into a TA Q50 thermogravimetric analyser (TA Instruments, New Castle, DE, USA). The residual mass of the sample was then continuously recorded (±0.1 mass

accuracy) as the sample was heated from ambient temperature to 950 °C at 10 °C·min−1 in an oxidative atmosphere (air) or



inert N2 atmosphere (both 50 mL·min−1 flow rate), inducing a process of thermal degradation under pyrolysis. All measured thermograms were evaluated using the TA Universal Analysis 2000 software, in which the derivation of the TG curves was also performed. DTG analysis of the tea samples was undertaken in the same manner as the soil samples, but using TA Q5000 thermogravimetric analyser (TA Instruments, New Castle, DE, USA). Prior to measurement, all samples were homogenised

by grinding in an agate mortar. For both soil and tea, the moisture content was determined from the relative weight loss corresponding to the minimum DTG curve, the residual mass at 950 °C representing the ash content (inorganic moieties).

### 2.4.2 UV-Vis spectrophotometry

UV-Vis spectrophotometry was used to measure polyphenolic and water-extractable compounds from organic matter in tea samples, according to the modified Folin-Ciocalteho method of Gao et al. (2019), using a Hitachi U3600H dual beam UV-Vis

spectrometer (Hitachi, Tokyo, Japan) with a wavelength range of 650–850 nm and gallic acid as a calibration solution. The content of all water-extractable polyphenolic compounds was measured in triplicate for each sample.

### 2.5 Statistical analysis

The difference between potential and actual soil reaction (pH) was expressed as $\Delta pH = pHKCl – pHwater$. The absolute difference between potential and actual H+ concentration was calculated by de-logarithming the pH values as $\Delta[H^+] = 10^{(-pHKCl)} – 10^{(-pHwater)}$.

Pairwise regression analysis was used to assess the significant difference between soil properties and selected environmental covariates in R v.4.2.2 (R Core Team 2022). All regression models were considered statistically significant at $p < 0.05$.

## 3. Results

### 3.1 Soil nutrient content

Analysis of nutrient content at each site revealed very low levels of exchangeable phosphorus (P), ranging from 0.0051–0.037 mg.g$^{-1}$, highly variable calcium (Ca) levels, ranging from 2.75–11.64 mg.g$^{-1}$, and magnesium (Mg) and potassium (K) levels ranging from 0.65–2.30 mg.g$^{-1}$ and 3.57–2.20 mg.g$^{-1}$, respectively (Table 2). The values of pHwater and pHKCl ranged from 6.48–8.17 (from slightly acidic to alkaline) and 5.53–6.67 (from acidic to neutral), respectively (Table 2). The storage acidity ($\Delta pH$) ranged from -0.74 – -1.79, the difference in hydrogen ion concentration ($\Delta[H^+]$) from $10^{(-5.57)} – 10^{(-6,71)}$ and the

coarse fraction content from 3.8–64.4% mass (Table 2).

**Table 2: Analysis of available nutrients (P = phosphorous, K = potassium, Ca = calcium, Mg = magnesium; mg.kg$^{-1}$; according to Mehlich 1984) and total nitrogen (Ntot; mg.kg$^{-1}$) content at each study site, along with percentage of coarse fraction. Site moisture conditions: *** = liquid water throughout the vegetation period, ** = variable dry periods, * =**



**dry site. pH values: pHwater = in distilled deionised water, pHKCl = in potassium chloride solution, ΔpH = difference between pHwater and pHKCl, Δ[H+] = converted to hydrogen ion concentration.**

| Site | Moisture | P | K | Ca | Mg | $N_{tot}$ | $pH_{water}$ | $pH_{KCl}$ | ΔpH | Δ[H$^+$] | Coarse fraction (% mass) |
|---|---|---|---|---|---|---|---|---|---|---|---|
| 1a | ** | 17.1 | 757 | 4537 | 956 | 200 | 7.22 | 5.78 | -1.4 | $10^{(-5.80)}$ | 23.3 |
| 1b | ** | 13.9 | 446 | 3710 | 754 | 400 | 6.98 | 5.87 | -1.1 | $10^{(-5.91)}$ | 13.6 |
| 2a | *** | 37.1 | 1417 | 4952 | 2340 | 6300 | 6.48 | 5.74 | -0.7 | $10^{(-5.83)}$ | 6.2 |
| 2b | **/*** | 35.9 | 1400 | 4735 | 2314 | 3600 | 6.54 | 5.76 | -0.8 | $10^{(-5.84)}$ | 3.8 |
| 3a | * | 20.2 | 703 | 3976 | 867 | 200 | 8.13 | 6.34 | -1.8 | $10^{(-6.35)}$ | 16.5 |
| 3b | * | 18.1 | 1258 | 3528 | 861 | 100 | 8.00 | 6.45 | -1.6 | $10^{(-6.46)}$ | 23.6 |
| 4 | **/*** | 8.8 | 2201 | 3418 | 1855 | 200 | 6.56 | 5.53 | -1,0 | $10^{(-5.57)}$ | 49.5 |
| 5 | *** | 14.9 | 354 | 11640 | 670 | 600 | 7.69 | 6.67 | -1,0 | $10^{(-6.71)}$ | 22.2 |
| 6 | *** | 14.9 | 882 | 4739 | 1652 | 100 | 7.29 | 5.57 | -1.7 | $10^{(-5.58)}$ | 25.1 |
| 7 | * | 10.1 | 394 | 3841 | 861 | 300 | 8.17 | 6.67 | -1.5 | $10^{(-6.68)}$ | 63.4 |
| 8 | * | 5.1 | 357 | 9687 | 649 | 200 | 8.03 | 6.67 | -1.4 | $10^{(-6.69)}$ | 12.8 |
| 9a | */** | 12.1 | 1000 | 2745 | 1007 | 100 | 7.82 | 6.07 | -1.8 | $10^{(-6.08)}$ | 57.9 |
| 9b | */** | 10.9 | 1185 | 3479 | 1255 | 300 | 7.30 | 5.85 | -1.5 | $10^{(-5.87)}$ | 53.2 |

## 3.2 Relationship between soil organic related fractions and environmental conditions

GRSP and EE-GRSP content were generally low, ranging from 0.096–0.968 mg.g$^{-1}$ and 0.072–0.752 mg.g$^{-1}$, respectively (Table 3). SOM content ranged from 2.79–15.04 wt.%, while SOC content ranged between 0.09–9.89 wt.%., though, with few

exceptions, most values were ≤ 1.5 wt.% (Table 3).

We recorded a very strong positive correlation between SOC and SOM (+0.93) and between GRSP and EE-GRSP (+0.91), and a strong correlation between EE-GRSP and SOC (+0.73). On the other hand, we recorded strong negative correlations between coarse fraction content and EE-GRSP (-0.67) and GRSP (-0.80), respectively, and between altitude and exchangeable P content (-0.71) (Figure 2).

## 3.3 Thermogravimetric and derivative thermogravimetric analysis of soil samples

Highest levels of soil loss (1.48–7.35 wt.%) were recorded in the lowest temperature range 200–430 °C, indicating that the majority of SOM fractions were relatively labile (Table 3). The mass loss in this temperature range was strongly correlated with both SOM (0.87) and SOC (0.73) content (Figure 2). Relatively high mass loss (0.77–2.76 wt.%) was also recorded at 430–590 °C, a range typically associated with loss of mineral fractions (i.e. clays) and organic matter associated with the clay

fraction. The lower levels of mass loss recorded at 590–750 °C (0.36–1.73 wt.%) are typically representative of the least labile



carbonates. Unexpectedly, the highest mass loss at Site 6 was recorded at 430–590 °C (1.84 wt.%), where N and SOM content are 0.1 wt.% and 1.4 wt.%, respectively (Table 3).

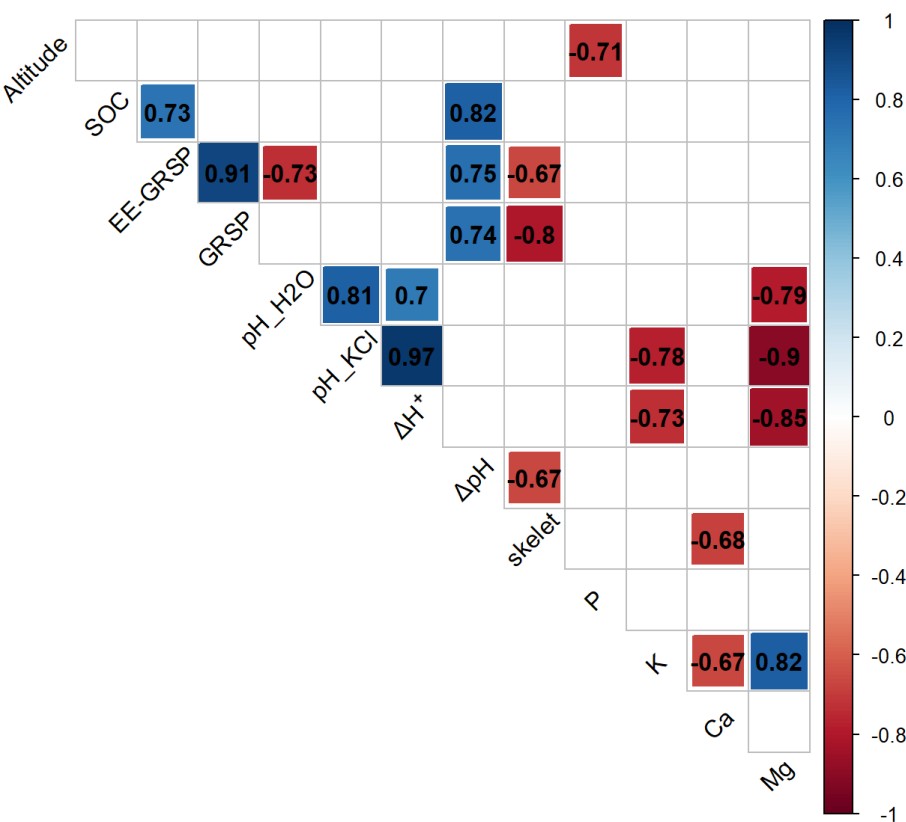

**Figure 2: Correlation diagram for selected environmental variables (altitude) and physical (skelet = coarse fraction content), chemical (SOC = soil organic carbon, EE-GRSP = easily extractable glomalin-related soil protein, GRSP = glomalin-related soil protein) and physicochemical (pH, exchangeable ion: K = potassium, Ca = calcium, Mg = magnesium, P = phosphorus contents) properties in Antarctic topsoil.**

**Table 3: Soil organic related fraction content, showing glomalin fractions (EE-GRSP = easily extractable glomalin-related soil protein, GRSP = glomalin-related soil protein), soil organic matter (SOM), soil organic carbon (SOC) and nitrogen (N; mean±SE) and mass loss (wt. %) during thermogravimetry over three temperature ranges, T1 = 200–430 °C, T2 = 430–590 °C, T3 = 590–750 °C.**

| Site | EE-GRSP (mg.g$^{-1}$) | GRSP (mg.g$^{-1}$) | SOC (mg.g$^{-1}$) | SOM (mg.g$^{-1}$) | N (mg.g$^{-1}$) | T1 (wt.%) | T2 (wt.%) | T3 (wt.%) |
|------|-----------|--------|---------|---------|---------|---------|---------|---------|
| 1a | 0.160 | 0.216 | 2.5 ±0.56 | 33.4 | 0.2 ±0.02 | 1.77 | 1.16 | 0.61 |





| | | | | | | | | |
|---|---|---|---|---|---|---|---|---|
| 1b | 0.088 | 0.560 | 4.7 ±0.09 | 38.0 | 0.4 ±0.01 | 2.05 | 1.08 | 0.55 |
| 2a | 0.752 | 0.968 | 98.9 ±3.31 | 143.8 | 6.3 ±0.28 | 7.35 | 2.76 | 1.73 |
| 2b | 0.640 | 0.904 | 45.1 ±1.15 | 150.4 | 3.6 ±0.05 | 6.70 | 1.70 | 1.21 |
| 3a | 0.112 | 0.176 | 2.4 ±0.05 | 32.2 | 0.2 ±0.01 | 1.60 | 1.07 | 0.51 |
| 3b | 0.080 | 0.192 | 0.9 ±0.02 | 27.9 | 0.1 ±0.00 | 1.48 | 0.99 | 0.41 |
| 4 | 0.552 | 0.672 | 31.2 ±1.49 | 65.5 | 0.2 ±0.01 | 3.31 | 1.08 | 0.79 |
| 5 | 0.128 | 0.192 | 5.9 ±0.13 | 41.7 | 0.6 ±0.00 | 2.32 | 1.16 | 0.65 |
| 6 | 0.104 | 0.136 | 1.4 ±0.07 | 35.5 | 0.1 ±0.01 | 1.75 | 1.84 | 1.29 |
| 7 | 0.072 | 0.096 | 11.5 ±1.11 | 46.6 | 0.3 ±0.01 | 2.03 | 1.34 | 0.90 |
| 8 | 0.216 | 0.760 | 14.6 ±0.74 | 40.7 | 0.2 ±0.02 | 1.66 | 1.05 | 0.69 |
| 9a | 0.088 | 0.120 | 1.1 ±0.08 | 31.1 | 0.1 ±0.01 | 1.70 | 0.77 | 0.36 |
| 9b | 0.176 | 0.200 | 3.9 ±0.14 | 38.7 | 0.3 ±0.00 | 1.89 | 1.00 | 0.51 |

**3.4 Thermogravimetric and derivative thermogravimetric analysis of standardised organic matter (tea) before and after decomposition.**

For all samples, DTGs of tea before decomposition showed a gradual decrease in mass as temperatures increased (Figs. 3, 4), with the most rapid loss coinciding with three peaks at 204, 300 and 350 °C, indicating loss of volatile compounds, hemicellulose and cellulose, respectively (Fig. 3). Peak mass loss during pyrolysis occurred at 300 °C, with a lesser peak occurring at 350 °C, though there was considerable overlap in the two peaks.

**Figure 3: Derivative thermogravimetric curve of a pre-decomposition tea sample during pyrolysis in an N2 atmosphere.**

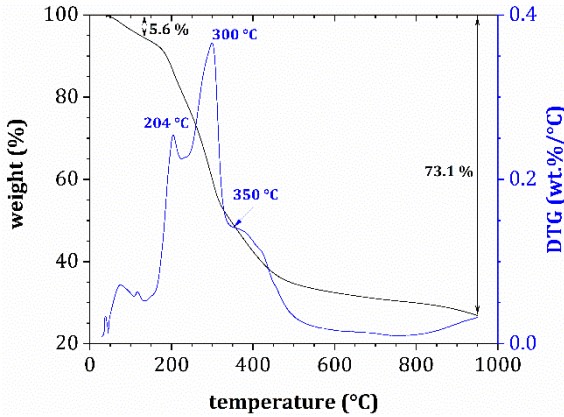



DTG plots for post-decomposition samples (Fig. 4) showed two main mass loss peaks at 320–351 °C and 396–416 °C. In all cases, the first peak found in pre-decomposition samples around 204 °C was either missing or much less pronounced. On the other hand, a less pronounced peak appeared sporadically in post-decomposition samples over the range 617–648 °C, which could be assigned to the highly stable substances such as lignin residues or humic substances in SOM. As such, post-decomposition OM showed higher thermal stability than the original samples, with the second and third mass loss peaks shifted

slightly toward higher temperatures.

**Figure 4: Post-decomposition tea sample derivative thermogravimetric curves (45 days exposure) during pyrolysis in an N2 atmosphere. Left = site 3a (seal cadaver); Right = site 6.**

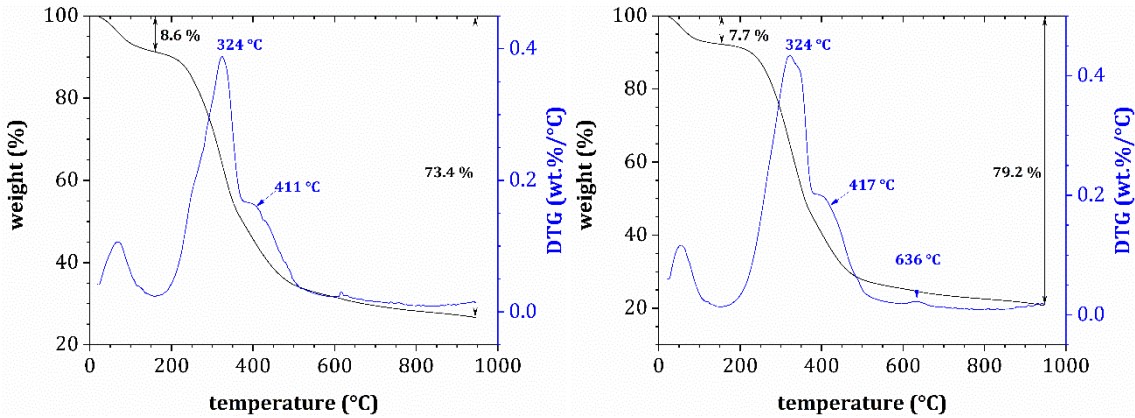

While N content increased at all sites post-decomposition (up to +2.6 wt.%), C content remained relatively stable, with only a slight increase or decrease (-1.01 – +1.3 %). As a result, the C:N ratio decreased from 12.5 to 7.1–10.2 post-decomposition, with the highest values recorded in the OTE at Site 9b. Overall, the C:N ratio tended to be lowest at lower altitude sites and sites supplied with water (see Tables 1, 4). In all cases, abundance of polyphenolic and water-extractable compounds decreased post-decomposition, while ash content decreased slightly at OTE sites (Sites 1b, 9b) but increased elsewhere, with the greatest

increase (+3.14 %) at site 2. At all sites, the amount of OM decreased by 2–6 % post-decomposition (Table 4).

**Table 4: thermogravimetric analysis of tea samples pre- and post-decomposition. Values for nitrogen (N) and carbon (C) = mean±SE, OM = organic matter.**

| Site | N (wt.%) | C (wt.%) | C:N ratio | Ash (wt.%) | OM (wt.%) | Polyphenols (mg·g$^{-1}$) | Water-extractable compounds (mg·g$^{-1}$) |
|------|----------|----------|-----------|------------|-----------|--------------------------|-------------------------------------------|
| | | | Before decomposition | | | | |
| | 3.69±0.06 | 46.18±0.66 | 12.5 | 4.02 | 90.70 | 78.78±8.54 | 8.88±1.04 |
| | | | After decomposition | | | | |





| 1b | 6.53±0.45 | 46.44±0.28 | 7.1 | 3.93 | 87.52 | 2.62±0.21 | 0.41±0.11 |
| 2a | 6.04±0.19 | 46.71±0.75 | 7.7 | 7.16 | 84.84 | 2.65±0.18 | 1.12±0.30 |
| 3a | 5.42±0.28 | 45.31±0.21 | 8.4 | 4.24 | 87.62 | 17.51±6.49 | 1.24±0.19 |
| 3b | 4.93±0.09 | 44.94±0.20 | 9.1 | 4.96 | 85.93 | 8.46±0.81 | 0.99±0.10 |
| 6 | 6.30±0.27 | 47.35±0.71 | 7.5 | 5.33 | 87.58 | 3.55±0.26 | 0.59±0.11 |
| 7 | 4.62±0.19 | 44.87±0.24 | 9.7 | 4.29 | 86.84 | 16.92±0.29 | 2.02±0.30 |
| 9b | 4.42±0.09 | 45.17±0.52 | 10.2 | 3.58 | 88.62 | 103.04±25.01 | 5.39±0.46 |

## 4. Discussion

### 4.1 Interaction between environmental conditions and available nutrients in fine-earth (Hypotheses H1 and H2).

We hypothesise that the highest GRSP and SOC content will be found in soils located at the lowest elevations, along with the lowest coarse fraction content (H1). We further hypothesise that GRSP content will be negatively correlated with content of exchangeable phosphorous (P) (H2)

The lack of any correlation between available nutrients and OM in our results suggest that the main factor determining nutrient distribution in Antarctic deglaciated soils is not presence of organic matter but rather environmental factors that accelerate mineral weathering processes. This is most likely due to the inhibition of biologically mediated nutrient cycling in the soil due to the extreme aridity and cold prevalent in Antarctica (McCraw, 1960; Campbell & Claridge, 1987). The important role of environmental factors, such as water availability and topography, is backed up by the positive correlation between available P and altitude (Fig. 2; thereby disproving Hypothesis H2). Unlike N, P lacks a biological fixation mechanism (Alewell et al., 2020). While sources of P are diverse, the primary source, particularly within Antarctic conditions, is through weathering of apatite (Filippelli & Souch, 1999), i.e. the inorganic 'pool' in which unweathered, mostly biologically-unavailable material, typically dominates (Cross & Schlesinger, 2001). A second major source, particularly in coastal areas, could be seabird droppings; however, P values recorded in previous studies (e.g. Zhu et al., 2014) suggest that levels in seabird droppings are around three orders of magnitude lower than those recorded in this study. As such, the correlation between available P and altitude is more likely to be indicative of a decrease in weathering intensity along the altitudinal gradient, with the intensity of weathering affected by the presence of running liquid water.

We forecast that ongoing warming from global climate change observed since the second half of 20th century will lead to an expansion in soil biodiversity and vegetation cover along the Antarctic weathering gradient due to an increase in mineral weathering intensity as levels of liquid water increase (Turner et al., 2005). This will further strengthen the relationship between fungal and photoautotrophic organisms, i.e. AMF, whereby the fungi give plants access to mineral-weathered P and the plants provide products of photosynthesis (i.e. C) to fungi. According to Blackwell (2000) and Simon (1993), this relationship has been in place since the earliest stages of terrestrial ecosystem colonisation.





## 4.2 Relationship between soil organic carbon and easily extractable glomalin-related soil protein (Hypothesis H3)

We further hypothesise that SOC will be positively correlated with GRSP content (H3). Under the Antarctic conditions, the

main sources of SOC are mosses (carbon = 40–50%; Green et al., 2008; Vingiani et al., 2004), lichens (genus Usnea; carbon = 25%; Shukla et al., 2019) and microorganisms associated with AMF (GRSP), with AMF hyphae containing around 50% carbon (Zhu & Miller, 2003) and glomalin (associated with AMF) containing levels range from 28 to 45% (Huang et al., 2011). In our case, however, we believe the main source of SOC was from glomalin (GRSP), as suggested by i) the strong correlation (r = 0.73) between EE-GRSP and SOC (thus confirming hypothesis H3), and ii) the relatively low conversion factor (0.36) for

the linear curve correlation between SOC and SOM (Table 3). In comparison, van Bemmelen (1889) reported a typical conversion factor of 0.58, while the UN FAO (2019) reports a common conversion factor of 0.43 to 0.58 (i.e. 43–58% SOC in SOM). This would suggest that the absence of any relationship between the glomalin fraction (GRSP) and mass loss over the first temperature interval during TGA (where OM fraction mass loss is expected) could have been caused by the lower C content in both glomalin fractions, i.e. lower than the commonly reported 58% C in OM (van Bemmelen, 1889).

The glomalin fraction content reported in this study (up to 0.97 mg.g$^{-1}$ GRSP; Table 2) was considerably higher than that reported in a previous study from James Ross Island, i.e. 0.01–0.15 mg.g$^{-1}$ (Pohanka & Vlček, 2018). This was most likely due to the study site of Pohanka & Vlček (2018) being situated close to the Whisky Glacier (Davis Dome), where the environmental conditions were unsuitable for vegetation. Surprisingly, the glomalin (GRSP) values in this study were actually higher than those recorded for agricultural soils, i.e. 0.30–0.70 mg.g$^{-1}$ (Wright & Anderson, 2000; Wuest et al., 2005). The high glomalin

values (GRSP) recorded in our study were the result of slow mineralisation of OM and its subsequent long-time accumulation. Glomalin (GRSP) clearly plays an important role in Antarctic soil genesis, not least due to its influence on buffering capacity. Correlations between $\Delta$ pH and SOC (0.82), $\Delta$ pH and GRSP (0.74) and $\Delta$ pH and EE-GRSP (0.75) all tend to indicate that fungi and their metabolites are contributing to the soil's buffering capacity; though the absence of any correlation between $\Delta$[H+] and SOC, EE-GRSP and GRSP does introduce some uncertainty.

As mentioned by Vlček et al. (2018), a SOC content of $\geq$ 2% is not unusual in Antarctica; however, it should be noted that Vlček et al. (2018) measured SOC was using the modified Walkley-Black method (1934), which can underestimate SOC values. Walkley (1947), for example, state that Walkley-Black recovery factors can range from 60 to 86%, while Grewal et al. (1991) reported recovery factors of 75 to 85%. In the present study, TGA/DTG values were comparable with those of Siewet (2004), who reported values ranging from 0.7 to 3.2% for 28 sites without vegetation on King George Island in the South

Shetlands.

## 4.3 Changes in the decomposition of standardised organic matter indicated by thermogravimetric analysis (Hypothesis H4)

We hypothesise a more rapid OM decomposition rate associated with increased soil SOC and GRSP content, and that this will be indicative of environments with increased microbial activity (H4). As expected, OM decomposition was intrinsically linked

with microbial activity (Fig. 4; Hypothesis H4). There was an apparent decrease in the abundance of polyphenolic and water-



extractable compounds in the post-decomposition tea sample, indicating that the most thermally-labile OM fractions (corresponding to a TGA peak at 204 °C in the original sample) had all been 'lost' during decomposition. Furthermore, the partially overlapping between the TGA and DTG curves in the incubated samples may indicate the formation of a complex organic compounds from labile fractions (which are characterised by a slightly bounded thermal stability).

Unexpectedly, we observed little change in total C post-decomposition but a noticeable decrease in the C:N ratio, suggesting that the tea sample had been colonised by N-fixing microorganisms (Utalie et al., 2021; Tóth et al., 2017; Kouchi et al., 2004). This could also help explain the relatively low C loss across sites, with any losses during mineralisation being compensated for by OM produced from microbial biomass (Paul; 2014; Keuskamp et al., 2013). This phenomenon is typical for the initial phase of mineralisation, where balance in the system has yet to be established (Golchin et al., 2018).

## 335 5. Conclusion

Most studies to date have assumed that climate, soil substrate and altitude are more important for Antarctic soil development than biological factors. In this study, we found that the main source of SOC in the deglaciated soils in James Ross Island, Antarctica, was the GRSP produced by microorganisms, an assumption supported by the low SOC:SOM conversion factor. During decomposition of a standardised OM sample (tea bag), the most labile OM elements (polyphenols and water-extractable

compounds) were lost relatively rapidly, after which the OM was colonised by N-fixing microorganisms, resulting in a decrease in the C:N ratio. The primary factor affecting nutrient accessibility in Antarctic soils proved not to be OM content but rather abiotic environmental factors that accelerate the weathering of minerals, e.g. availability of liquid water. Nevertheless, OM had a significant buffering function where GRSP was important in the development of polar soils. This was supported by the lack of any relationship between available nutrients and OM, most likely a result of inhibition of biologically mediated nutrient

cycling in the soil due to the region's extreme aridity and cold climate.

**Competing interests**

The contact author has declared that none of the authors has any competing interests.

**Acknowledgement**

The authors would like to thank to all the members of at Masaryk University based Czech Antarctic Research Programme for providing data and for fieldwork support.

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
