# Peer review of "Soil organic matter interactions along the elevation gradient of the James Ross Island (Antarctica)"

_EGUsphere, 2024_

## Author Comment (AC2)

```
############################################################
Analysis of Variance Table

Response: log10(P+1)
                  Df  Sum Sq  Mean_Sq  F_value  Pr(>F)
log10(Altitude+1)  1  0.17707 0.177071  6.1959  0.04165 *
Residuals          7  0.20005 0.028579
* * *
Signif. codes:  0 '***' 0.001 '**' 0.01 '*' 0.05 '.' 0.1 ' ' 1

Multiple R-squared:  0.4695,   Adjusted R-squared:  0.3938

ASSESSMENT OF THE LINEAR MODEL ASSUMPTIONS
USING THE GLOBAL TEST ON 4 DEGREES-OF-FREEDOM:
Level of Significance =  0.05

                   Value p-value                  Decision
Global Stat       0.76230  0.9434 Assumptions acceptable.
Skewness          0.01771  0.8941 Assumptions acceptable.
Kurtosis          0.52677  0.4680 Assumptions acceptable.
Link Function     0.18207  0.6696 Assumptions acceptable.
Heteroscedasticity 0.03575 0.8500 Assumptions acceptable.
```

Phosphorus vs. altitude

[Figure]

```
############################################################
Analysis of Variance Table

Response: log10(SOC+1)
          Df  Sum Sq Mean Sq F value   Pr(>F)
ΔpH        1 1.48086 1.48086  14.629 0.006502 **
Residuals  7 0.70859 0.10123
* * *
Signif. codes:  0 '***' 0.001 '**' 0.01 '*' 0.05 '.' 0.1 ' ' 1

Multiple R-squared:  0.6764,   Adjusted R-squared:  0.6301

ASSESSMENT OF THE LINEAR MODEL ASSUMPTIONS
USING THE GLOBAL TEST ON 4 DEGREES-OF-FREEDOM:
Level of Significance =  0.05

                        Value p-value                    Decision
Global Stat          0.642930  0.9582 Assumptions acceptable.
Skewness             0.009467  0.9225 Assumptions acceptable.
Kurtosis             0.496774  0.4809 Assumptions acceptable.
Link Function        0.065113  0.7986 Assumptions acceptable.
Heteroscedasticity   0.071576  0.7891 Assumptions acceptable.
```

SOC vs. ΔpH

[Figure]

```
###########################################################
Analysis of Variance Table

Response: GRSP
             Df Sum Sq Mean Sq F value   Pr(>F)
log10(SOC+1)  1 545470  545470  14.562 0.006576 **
Residuals     7 262210   37459
* * *
Signif. codes:  0 '***' 0.001 '**' 0.01 '*' 0.05 '.' 0.1 ' ' 1

Multiple R-squared:  0.6754,   Adjusted R-squared:  0.629

ASSESSMENT OF THE LINEAR MODEL ASSUMPTIONS
USING THE GLOBAL TEST ON 4 DEGREES-OF-FREEDOM:
Level of Significance =  0.05

                        Value p-value                  Decision
Global Stat           5.25450  0.2622 Assumptions acceptable.
Skewness              1.36409  0.2428 Assumptions acceptable.
Kurtosis              0.04924  0.8244 Assumptions acceptable.
Link Function         2.50859  0.1132 Assumptions acceptable.
Heteroscedasticity    1.33258  0.2483 Assumptions acceptable.
```

GRSP vs. SOC

---

## Author Response (AR1)

**Response to reviewers**

Dear editors

I would like to extend my sincere gratitude to the reviewers for their insightful comments and constructive feedback. Your suggestions have significantly contributed to improving the quality and clarity of this work. I deeply appreciate the time and effort you invested in reviewing this manuscript, and your valuable insights have been instrumental in enhancing its overall contribution to the field.

*Reviewer comments in italic*

**CC1:** *This study is very thorough. However, my recommendation would be the addition of the date of sampling. This is missing in the manuscript, and it could potentially be very useful for future studies to be done by other authors.*

Authors: The date of sampling was added to the methods chapter.

**CC2:** *This manuscript examines SOM interactions on James Ross Island, Antarctica, focusing on SOC, GRSP, P, and microbial activities across altitudes. It provides insightful data on how extreme polar conditions affect soil C dynamics, contributing to our understanding of carbon sequestration processes in polar ecosystems. Despite its promising contributions, the manuscript would benefit from further refinement in areas such as methodological descriptions, statistical analyses, and discussion to enhance overall clarity and impact. The introduction of this manuscript could benefit from a clearer connection between global soil C significance and the specific contexts of Antarctic soils. Currently, the introductory section opens with a sentence that seems unrelated to the central theme of the research (L39), which might distract from the focal point of the study. By consolidating the first and second paragraphs into a more cohesive unit, the narrative could be strengthened, thus providing a clearer justification for the research focus.*

**Authors response:** the potentially problematic parts have been identified and better linked each other.

*Materials and Methods:*

1. *Please verify and standardize subscript and superscript usage throughout this section to maintain consistency.*

**Authors response:** it has been corrected.

2. *It is essential to specify the number of soil samples collected at each site. Detailing this information will strengthen the study's replicability and statistical validation.*

**Authors response:** detailed description of sampling method and number of samples has been added.

3. *Section 2.3 lacks details on soil C and N testing methodologies.*

**Authors response:** the analyses of C and N were performed by the thermogravimetry. The details of testing methodologies are presented in chapter 2.4.1 (in Materials and Methods).

4. *The absence of ANOVA or similar statistical tests to analyze data variability among elevation gradients and treatments should be addressed.*

**Authors response:** Based on the technical limitations to clearly selected groups of variables to categorize the data without the strong influence of climate and soil factor we decided to cover the heterogeneity of environment as much as possible and use the correlation tests. This design doesn't allow divided data into the groups and use ANOVA or similar statistical tests. It seems to be a good decision and provided promising results, see Figure A1.

[Figure]

Figure A1 Correlation of P and altitude

5. *The inclusion of delta pH in the statistical analysis needs reevaluation.*
6. *I have not seen papers using the difference between $pH_{H2O}$ and $pH_{KCl}$ as an indicator of soil buffering capacity. I am not saying that's incorrect, but an explanatory experiment or references justifying this approach would be beneficial.*
7. *Please use built-in functions for equations like this ($\Delta[H+] = 10^{(-pHKCl)} – 10^{(-pHwater)}$) to enhance readability.*

**Authors response to points 5+6+7:** The paragraph with appropriate reference has been added to M&Ms (chapter 2.3) for better understanding the role and importance of delta pH in the local carbon cycle interpretation.

*Results:*

8. *Inclusion of statistical significance and standard error details in Tables 2-4 is necessary to substantiate the research findings.*

**Authors response:** the standard error details have been added. The correlation coefficient is presented in the Figure 2 where just the statistically significant correlations ($P<0.05$) are presented. "Only significant correlations at $P<0.05$ are shown" was add to description of Figure 2.

9. *For Table 2:*
   a. *the symbols (\*\*\*, \*\*, \*) generally used to denote statistical significance. Please consider used other symbols to avoid any potential confusion;*

**Authors response:** the symbols were replaced by abbreviations "w", "sd" and "d" for wet, semi-dry and dry moisture conditions respectively.

   b. *The term "coarse fractions" needs to be specified—do these refer to gravels or sands?*

**Authors response**: coarse fraction is related to soil texture classes. Coarse fraction covers the particles larger than 2 mm. Description "coarse fractions" in the Table 2 was change to Texture class > 2 mm.

   c. *M&M does not address the presence of total N explicitly. If total N is measured, does and how SOC is measured? In addition, I don't really understand why SOC is in Table 3 instead of Table 2. I think Table 2 and some parts of Table 3 should be combined or represent their data through a figure that illustrates changes across elevation gradients.*

**Authors response**: The nitrogen was measured by thermogravimetry same as SOC. The N is placed in Table 3 because we would like to present all the data from thermogravimetry in one table. The sentence "Soil properties analysed thermogravimetry." was added to the description of Table 3. The relationship between the soil data and elevation is clearly presented in Figure 2. Figure of soil data changes across elevation gradients (Figure A1) will show the same as the Figure 2 and would cause the results duplication.

*d. The rationale for displaying both ∆pH and ∆[H+] is unclear. The expression 10^(-5.80) may not be readily interpretable by all readers.*

**Authors response**: The paragraph with appropriate reference has been added to M&Ms (chapter 2.3) for better understanding the role and importance of delta pH in the local carbon cycle interpretation.

10. *Table 3: Please specify what "N" stands for—is it referring to soil total N?*

**Authors response**: N is referring to total nitrogen content. It has been specified in the manuscript.

11. *Figures 3 and 4: captions should be positioned underneath the figures.*

**Authors response**: it has been corrected.

*Discussion:*

12. *L271-273 redundantly states the last part of Introduction.*

**Authors response:** the problematic part has been removed.

13. *L286: I think another important thing that could be mentioned is P has low mobility. Apatite has calcium. If the decrease in weathering intensity of apatite is indeed the cause behind the correlation observed between Melich-3-P and altitude, then why is it that we do not observe a similar correlation between altitude and available Ca?*

**Authors response:** Calcium is one of the most common elements on Earth and is contained in many major rock-forming minerals, such as silicates and carbonates (e.g., feldspars, calcite etc.). The concentration of 3,000-11,000 mg/kg of Ca found in the soil samples cannot originate exclusively from apatite weathering; there could be various sources of calcium, and therefore, a correlation with altitude cannot be expected. Additionally, Ca is generally more mobile in soils than P because it remains in the soil solution as a free ion and does not bind as strongly to soil particles; therefore, the correlation between calcium and altitude may not be representative.

14. *While the manuscript provides data on total N, the discussion on N cycling is notably brief. Given the unique environmental conditions of Antarctica, incorporating a detailed analysis of N transformations could significantly enhance the understanding of nutrient dynamics in polar ecosystems. The effects of Antarctic conditions, such as low SOM concentrations and C/N inputs, along with the ecological influences of freeze-thaw cycles and seabird activities, warrant deeper exploration.*

**Authors response:** We acknowledge the general potential of N in the environment. However, due to the lack of correlation between nitrogen and altitude, as well as its generally low concentration in the soils, we focused primarily on phosphorus (P), where we found a link with altitude.

15. *Something about Table 3:*
    a. *I didn't find any linear curve in Table 3 (L300).*
    b. *The most important data in this study were in Tables 2-4 and Figure 2. I do think the authors have spaces to provide more in-depth relationship of measured parameters, like presenting the linear curve mentioned in L300.*

**Authors response:** we apologize, it was a misunderstanding the text has been corrected.

    c. *By just looking at numerical values in Table 3, I felt the deviation of the conversion factor would be large. The authors should also provide $R^2$ and p-value of the regression lines.*

**Authors response:** it is not relevant to Table 3. The standard deviation has been added. The correlation coefficient is presented in the Figure 2 where just the statistically significant correlations (P<0.05) are presented. "Only significant correlations at P<0.05 are shown" was add to description of Figure 2.

    d. *Another question is, is C in GRSP not part of SOC? The authors didn't provide information about C (and SOM) analysis in M&Ms, but I think GRSP is part of SOC for dry combustion analysis. If C analysis approach in this study is not the most common approach, does it make sense to compare the conversion factor to studies with different SOC/SOM measurement method?*

**Authors response:** Both GRSP and EEGRSP are part of soil organic matter (SOM). However, we wanted to determine, by correlating with other properties, how much of a substantial part of SOM it is. The SOM content (in thermogravimetry analysis) was calculated by the difference between total mass and non-combustible share, and the data obtained were corrected for moisture content. The explanation was added to the M&Ms (chapter 2.4.1).

16. *I am not sure how to interpret Fig 4 to get this statement "OM decomposition was linked with microbial activity".*

**Authors response:** Some of the changes observed in the TBI analysis can be (at least partially) attributed to thaw/freeze cycles or the accessibility of liquid water, which reduces water-extractable compounds. However, we assume that the most significant changes have occurred due to microbial decomposition, as indicated by changes in polyphenol content, an increase in nitrogen content, and other factors. This issue is described in chapter 4.3.

**CC3:**
*1. Could you please try to combine Results and Discussion into one chapter, so it will*

*be easier for readers to follow the discussion after the respective results (which are many and of great importance for soil and permafrost community and not only for them!)*

**Authors response:** Thank you for the suggestion. We recognize the potential benefits of combining the Results and Discussion into a single chapter to enhance readability. However, to maintain clarity and focus, especially considering the significance of the results, we have chosen to keep them separate. This structure allows for a clear presentation of the data, followed by a comprehensive analysis.

*2. Please think about performing (if possible) some detailed SOM molecular study, i.e. 13C-NMR or FTIR, to show the difference in molecular composition in soil organic matter along the studied environmental gradient. This is also imprtant for assessment of stability of organic matter and for assessment of current and potential stocks of organic carbon and related to the degree of humification in Antarctic soils.*

**Authors response**: We acknowledge the value of conducting detailed studies SOM at the molecular level, such as using 13C-NMR or FTIR, to gain a deeper understanding of the molecular composition and stability of organic matter across environmental gradients. However, due to current resource and logistical constraints, we were unable to perform these analyses in this study. Our primary focus is on evaluating the influence of environmental factors on the distribution of nutrients and organic matter in the environment. A detailed analysis of the distribution of individual fractions of OM in Antarctic soils, based on the significant environmental factors identified in this study, could be a focus of future research.

*3. Please try to expand the "Conclusions" part, since in the present form there are less conclusions than results provided in the manuscript (which are of course worth to be explained as Conclsuions of the presented work).*

**Authors response**: conclusions have been improved according to recommendations.

---

## Author Response (AR2)

**Author's Response**

The changes to the manuscript *egusphere-2024-607-manuscript-version2.pdf* have been approved as part of the recent minor revisions. No new text has been added to the manuscript, except for the acknowledgements, which mention the project's funding for the article's publication.

Vitezslav Vlcek